# Intranasal Nanotransferosomal Gel for Quercetin Brain Targeting: I. Optimization, Characterization, Brain Localization, and Cytotoxic Studies

**DOI:** 10.3390/pharmaceutics15071805

**Published:** 2023-06-23

**Authors:** Mohammed H. Elkomy, Randa Mohammed Zaki, Omar A. Alsaidan, Mohammed Elmowafy, Ameeduzzafar Zafar, Khaled Shalaby, Mohamed A. Abdelgawad, Fatma I. Abo El-Ela, Mostafa E. Rateb, Ibrahim A. Naguib, Hussein M. Eid

**Affiliations:** 1Department of Pharmaceutics, College of Pharmacy, Jouf University, Sakaka 72341, Saudi Arabia; osaidan@ju.edu.sa (O.A.A.); melmowafy@ju.edu.sa (M.E.); azafar@ju.edu.sa (A.Z.); khshalabi@ju.edu.sa (K.S.); 2Department of Pharmaceutics, College of Pharmacy, Prince Sattam Bin Abdulaziz University, Al-Kharj 11942, Saudi Arabia; randazaki439@yahoo.com; 3Department of Pharmaceutics and Industrial Pharmacy, Faculty of Pharmacy, Beni-Suef University, Beni-Suef 62511, Egypt; 4Department of Pharmaceutical Chemistry, College of Pharmacy, Jouf University, Sakaka 72341, Saudi Arabia; mhmdgwd@ju.edu.sa; 5Department of Pharmacology, Faculty of Veterinary Medicine, Beni-Suef University, Beni-Suef 62511, Egypt; fatma.aboel3la@vet.bsu.edu.eg; 6School of Computing, Engineering & Physical Sciences, University of the West of Scotland, Paisley PA1 2BE, UK; mostafa.rateb@uws.ac.uk; 7Department of Pharmaceutical Chemistry, College of Pharmacy, Taif University, Taif 21944, Saudi Arabia; i.abdelaal@tu.edu.sa

**Keywords:** quercetin, transferosomes, brain targeting, CT scan, cytotoxicity, intranasal

## Abstract

Numerous neurological disorders have a pathophysiology that involves an increase in free radical production in the brain. Quercetin (QER) is a nutraceutical compound that shields the brain against oxidative stress-induced neurodegeneration. Nonetheless, its low oral bioavailability diminishes brain delivery. Therefore, the current study aimed to formulate QER-loaded transferosomal nanovesicles (QER-TFS) in situ gel for QER brain delivery via the intranasal route. This study explored the impacts of lipid amount, edge activator (EA) amount, and EA type on vesicle diameter, entrapment, and cumulative amount permeated through nasal mucosa (24 h). The optimum formulation was then integrated into a thermosensitive gel after its physical and morphological characteristics were assessed. Assessments of the optimized QER-TFS showed nanometric vesicles (171.4 ± 3.4 nm) with spherical shapes and adequate entrapment efficiency (78.2 ± 2.8%). The results of short-term stability and high zeta potential value (−32.6 ± 1.4 mV) of QER-TFS confirmed their high stability. Compared with the QER solution, the optimized QER-TFS in situ gel formulation exhibited sustained release behavior and augmented nasal mucosa permeability. CT scanning of rat brains demonstrated the buildup of gold nanoparticles (GNPs) in the brains of all treatment groups, with a greater level of GNPs noted in the rats given the transferosomal gel. Additionally, in vitro studies on PCS-200-014 cells revealed minimal cytotoxicity of QER-TFS in situ gel. Based on these results, the developed transferosomal nanovesicles may be a suitable nanocarrier for QER brain targeting through the intranasal route.

## 1. Introduction

Numerous neurological disorders have a pathophysiology that involves an increase in free radical production in the brain [1]. Through the microbiome-gut-brain axis, recent evidence suggests that the microbiome in the gut may affect the brain [2]. The dysfunction of this axis may be crucial in the etiology of neurological disorders [3]. This mechanism includes CNS, the enteric nervous system, the intestinal mucosal barrier, the immune system, various neurotransmitters, neuroregulators, and the blood–brain barrier (BBB) [4].

Presently, phytotherapy, which emphasizes using plants to treat illness [5,6,7], is a valuable resource for developing more effective therapeutic agents. Various medicinal plants have been employed to treat various neurological illnesses [8,9]. Due to their antioxidant and neuroprotective effects [10], flavonoids are a potential class of natural substances with a significant role in neuropharmacology. These bioactive substances are abundant in fruits and vegetables [11].

Quercetin (QER) is the flavonoid found in most food and is mainly utilized by people [12]. Abundant evidence has referred that fruits and vegetables shield the brain against oxidative stress-induced neurodegeneration because of their high QER concentration [13,14]. By suppressing hydroperoxide production, restoring antioxidant enzymes, and decreasing free radicals, QER protects neurons from oxidative stress [15,16]. However, QER has low distribution potential to the brain after oral delivery owing to difficulties in crossing BBB [17] as well as to rapid metabolism [18]. Therefore, QER exerts its neuroprotective effects several hours after delivery [19]. It has also been demonstrated to have anxiolytic and cognitive-enhancing properties [20]. Several studies demonstrate the preventive function of QER against depressive-like illnesses [21,22]. In addition, this flavonoid ameliorates anxiety and depression caused by the corticotropin-releasing factor in mice and rats [23,24]. Despite extensive in vivo studies, the potential of QER is not fully explored pertaining to inherited challenges, such as low oral bioavailability, hepatic metabolism, physiological pH instability, low brain permeation, photodegradation, and hydrophobic nature [25,26,27]. Therefore, a suitable nanocarrier is necessary to overcome the obstacles mentioned above. 

The nose-to-Brain (NTB) route has received tremendous attention lately as a viable route as it offers various benefits over oral or parenteral delivery [28,29]. In addition to being non-invasive and painless, the NTB route has a high level of vascularization via the epithelium, which increases medication absorption [30]. In addition, the NTB route bypasses hepatic and intestinal metabolism and permits medication transport to the brain indirectly via systemic circulation and directly through the olfactory and trigeminal neurons [28,31]. Recent research indicates that the NTB route may be beneficial in the treatment of several CNS disorders, including brain tumors [32], schizophrenia [33], and Alzheimer’s disease [29]. Nevertheless, the limited surface area of the nasal cavity, quick mucociliary clearance, and limited mucosal permeability are believed to be the principal constraints of the NTB route [28,29,30]. Consequently, most contemporary research focuses on producing mucoadhesive nasal formulations and using permeation enhancers to boost residence duration and nasal uptake [30].

Recent research has focused on vesicular nanocarriers, such as liposomes [34], cubosomes [28], transferosomes [30], and niosomes [35], to deliver pharmaceuticals to the brain via the intranasal (IN) pathway. Nanovesicular systems can carry drugs from the intranasal mucosa to the systemic circulation and brain due to their incredible penetrating properties and capacity to ameliorate ciliary clearance [30]. Transferosomes (TFS), one of these vesicular carriers, are being studied extensively and have become more significant lately because they can transport medications to the brain through intranasal administration [30]. Transferosomal nanovesicles are more deformable than conventional liposomes [36]; therefore, they are called ultra-deformable liposomes. TFS is highly adaptable, elastic in nature, and stress-responsive [36]. Other than that, it boosts drug molecule permeation by squeezing the vesicle structure between cells [30] and facilitates their penetration via holes much smaller than their size. Transferosomal infrastructure can house both hydrophobic and hydrophilic moieties [36]. Recently, TFS has been investigated as a carrier for brain delivery for various pharmaceuticals, such as resveratrol [37], olanzapine [38], and zolmitriptan [39]. 

This study aimed to explore the potential of QER-loaded transferosomal nanovesicles (QER-TFS) to deliver QER to the brain through the intranasal route. In this study, QER-TFS were developed, optimized, and assessed with respect to their physical and morphological characteristics to establish their appropriateness for intranasal application. Next, the optimized formulation was integrated into a thermosensitive gel for brain localization and cytotoxic studies. In a subsequent study, the effect of intranasal QER-TFS gel on antidepressant activity was investigated in a depressed rat model.

## 2. Materials and Methods

### 2.1. Materials

Quercetin (QER), Dialysis membranes (MW cutoff: 12 kDa), Sodium deoxycholate (SDC), Methanol (HPLC grade), MTT, L-α-Phosphatidylcholine (Lecithin), Chloroform (HPLC grade), and Dimethyl sulfoxide (DMSO) were procured from Sigma-Aldrich (St. Louis, MO, USA). Gentamycin, RPMI-1640, and Fetal Bovine serum were purchased from Lonza (Verviers, Belgium). PCS-200-014 cells (human oral epithelial cell line) were obtained from the American Type Culture Collection (Rockville, MD, USA). Other chemicals and solvents used were of analytical grade.

### 2.2. Methods 

#### 2.2.1. Design and Optimization of Experiments

A central composite design (CCD, face-centered), one categorical factor with two levels (−1, +1), and two numeric factors with three levels (−1, 0, +1) were used to optimize the independent variables for the formulation of QER-TFS. The optimization algorithm was set to decrease vesicle size (VS) and increase entrapment efficiency (EE%), and the cumulative amount permeated through nasal mucosa after 24 h (Q_24_). Design Expert^®^ (version 12.0.3.0, USA) was used to construct and evaluate the experimental design, and the optimum formulation was chosen. Furthermore, the statistical significance of each model coefficient was assessed using the multi-factor ANOVA test implemented in the Design Expert software. The lack-of-fit test was also used to assess the significance of the model, and the model was deemed acceptable if the variance due to lack of fit did not differ significantly from the variance due to pure error. The selection of the best models is based on several criteria (e.g., model *p*-value, lack of fit *p*-value, adjusted R^2^, and predicted R^2^). There were 20 trials, with 16 referring to the centroid of each tridimensional cube edge and 4 to the center point replicas. Three independent variables were tested: Lipid amount (X_1_), EA amount (X_2_), and EA type (X_3_). VS (nm), EE (%), and Q_24_ (%) were the response variables considered. We obtained 3D response surface plots using the plot3D R-package [40,41]. The experimental runs of the QER-TFS as per the CCD design are shown in Table 1. 

#### 2.2.2. Formulation of QER-TFS

QER-TFS were formulated via the thin film hydration technique [30]. In a round-bottom flask, QER (10 mg), Lecithin, and EAs were dissolved in a 15 mL chloroform and methanol (2:1) solution. The chloroform/methanol mixture was vacuum vaporized in a rotating evaporator (Stuart rotary evaporator, UK) at 60 °C to enable the creation of a thin film and then maintained under vacuum (2 h) to ensure that the organic solvents were completely evaporated. Then, phosphate-buffered saline (PBS, pH 7.4, 10 mL) was added to the round-bottom flask. The dried film was then completely hydrated with rotation at 60 rpm (25 °C, 1 h). The dispersion produced was sonicated for 10 min and refrigerated overnight before characterization.

#### 2.2.3. Characterization and Optimization of QER-TFS

##### Vesicle Size Analysis

The VS of QER-TFS was established via dynamic light scattering approaches (Zetasizer Nano ZS, Malvern Instruments, Malvern, UK) [41,42]. Prior to the assessment, QER-TFS dispersion was diluted with purified water (1:10) and investigated at 25 °C [43].

##### Entrapment Efficiency 

The entrapment of QER in QER-TFS was estimated indirectly by measuring the QER in the supernatant following centrifugation and subtracting this amount from the amount used during formulation. The supernatant containing free QER was separated using a cooling centrifuge (SIGMA 3-30K, Roedermark, Germany) at 15,000 rpm (4 °C, 2 h) [44]. Using a spectrophotometer (SHIMADZU (UV-160A), Tokyo, Japan) at 375 nm, the concentration of QER in the supernatant was measured after an appropriate dilution [45]. The EE (%) of QER was assessed using the following formula [40,46]:$$EE\;\%=\frac{Total\;QER\;added-QER\;in\;Supernatant}{Total\;QER\;added}\times 100$$

##### Ex Vivo Permeability 

In a Franz Diffusion cell (2.5 cm^2^), an ex vivo nasal diffusion experiment was conducted using nasal mucosa removed from freshly sacrificed sheep. After extensively exposing the nasal septum and detaching the nose from the cranium, the nasal mucosa was removed carefully with forceps and surgical scissors. The mucosal tissues were submerged for 30 min in PBS (pH 6.5) [29]. Fifty milliliters of PBS with Tween 80 (0.01% *v/v*) were added to the receptor compartment at a temperature of 37 °C with stirring at 50 rpm [28]. Then, freshly excised sheep nasal mucosa (0.2 mm thickness) was carefully placed between the donor and receptor vessels. Before applying different QER-TFS formulations, each comprising 3 mg of QER, to the mucosal surface of the nasal mucosa, the nasal mucosa was kept for 15 min to equilibrate. After 24 h, the aliquots were taken from the receptor compartment. Each sample was centrifuged, filtered, and spectrophotometrically analyzed for its QER concentration. 

#### 2.2.4. Characterization of Optimized QER-TFS

##### Fourier-Transform Infrared Spectroscopy (FT-IR)

The QER and TFS matrix interaction was assessed using FT-IR. Potassium bromide (KBr) was blended in a mortar and pestle with QER, lecithin, SDC, empty TFS (no drug), and QER-TFS. The blends were then compressed into a thin film using a hydraulic press and placed in sample holders. Then, using KBr film as a standard, FT-IR analysis was carried out in transmission mode between 4000 and 400 cm^−1^ [41].

##### Differential Scanning Calorimetry (DSC)

For further investigation into the QER/TFS matrix interaction, DSC thermograms were recorded using a calorimeter (DSC-60, Shimadzu, Kyoto, Japan) for QER, lecithin, SDC, empty TFS (no drug), and QER-TFS. Seven milligrams of samples were accurately weighed, placed in aluminum pans, and tested using an empty aluminum pan as a standard [41]. The samples were heated to 300 °C at 10 °C/min while being scanned in a nitrogen atmosphere.

##### Morphology and Zeta Potential Analysis

The optimum QER-TFS formulation was scrutinized for its morphological merits using transmission electron microscopy (TEM). One drop of freshly manufactured QER-TFS dispersion was put into a carbon-coated grid and kept at 25 °C (5 min) after sufficient dilution [47]. The grid was then stained negatively with phosphotungstic acid (1%, *w/v*), let to settle to allow for proper staining absorption, and then the stain was removed using filter paper. At 80 kV, the formulation was examined by TEM (Jeol, Tokyo, Japan) [44]. The zeta potential (ZP) of the optimized QER-TFS was estimated with the same equipment for VS analysis. 

#### 2.2.5. Preparation of QER-TFS In Situ Gel

The optimal formulation was incorporated into a mucoadhesive in situ gel formulation to facilitate intranasal delivery, extend residence time, and enhance absorption. The optimum QER-TFS and QER solution (QER-SOLN) were incorporated into a gel base (Carbopol 971P (0.5%), Poloxamer 188 (10%), and Poloxamer 407 (20%)) using the cold technique [30].

#### 2.2.6. In Vitro Release

The Vertical Franz cells were used for the drug release experiments (2.5 cm^2^ diffusion surface area). Before testing, a dialysis membrane was submerged for 24 h in a simulated nasal electrolyte solution (SNES) before being positioned between the donor and receptor chambers. QER-TFS gel and QER-SOLN gel with equivalent QER amounts (3 mg) were introduced to the donor chamber. The experiment was conducted at 37 °C with a medium volume of 50 mL SNES containing Tween 80 (0.01% *v*/*v*) and stirred at 50 rpm [30,48]. At predetermined intervals, 1 mL of the receptor medium aliquots were removed and replaced with 1 mL of fresh SNES to maintain a constant volume. The samples were filtered, and the cumulative quantity released in the collected samples at certain time intervals was determined spectrophotometrically. The relationship between the percentage of drug released and time was determined.

#### 2.2.7. Ex Vivo Permeability

Using the Franz diffusion cell, as previously mentioned, ex vivo diffusion tests of QER via sheep nasal mucosa were carried out on the optimized QER-TFS gel and QER-SOLN gel. Permeation parameters for both QER-TFS gel and QER-SOLN gel were calculated using a previously published approach [43].

#### 2.2.8. Stability Study

The optimized QER-TFS gel was kept for 90 days in a glass jar (4 °C). After a storage period of 0, 30, 60, and 90 days, the VS, EE, and ZP of samples from the optimal formulation were evaluated [43]. In addition, frequent visual inspections were conducted to identify any indications of physical instability, such as aggregations, separation, or precipitation.

#### 2.2.9. Evaluation of pH

Measuring the pH of the optimal QER-TFS gel is necessary to ensure it does not irritate the nasal mucosa following administration. The pH was estimated via a digital pH meter (Jenway, UK).

#### 2.2.10. Determination of QER Localization in the Brain

The brain accumulation of QER-TFS capped with gold nanoparticles (GNPs) was examined via Computed Tomography (CT) Scanning using twelve healthy adult Wistar rats weighing between 190 and 210 gm. The accommodation for the animals was well-ventilated, and the animals had unrestricted access to food and water. The animals were given seven days to acclimate prior to the start of the experiment.

##### Preparation of QER-TFS Capped with GNPs (QER-TFS-GNPs)

GNPs were prepared using a reduction process, as previously stated [49]. In a nutshell, a boiling solution of gold chloride was mixed with trisodium citrate and agitated for around 4 min to produce a stunning wine-red color [50]. The optimized QER-TFS formulations were then capped with the developed GNPs during the hydration process [51], and they were subsequently included in the previously mentioned thermosensitive gel.

##### CT Scanning

It has been reported that GNPs can be identified in vivo by CT scanning due to the fact that they are exceptional contrast agents and induce significant X-ray attenuation due to their high atomic number; therefore, GNPs can be used to identify targeted tissues [52,53]. CT scanning was used to evaluate the accumulation of QER-GNPs (intranasal and oral) and QER-TFS-GNPs (intranasal) in the separated brains. The rats were divided into four groups (n = 3): the control group (untreated), the intranasal QER-GNPs gel-treated rat group, the oral QER-GNPs treated rat group, and the intranasal QER-TFS-GNPs gel-treated rat group. The respective groups received intranasal QER-GNPs gel or QER-TFS- GNPs gel at a QER dose of 10 mg/kg and oral QER-GNPs at a dose of 300 mg/kg of QER. The rats were killed two hours after the drug was delivered, and the extracted brains were saline-washed before being fixed in 10% formalin. The separated brains were put in a glass dish and scanned using a CT scanner (Sensation 64, Siemens CO, Erlangen, Germany) [53]. The accumulation and cellular absorption of GNPs in the brains were investigated and compared to the control group.

#### 2.2.11. In Vitro Assessment of Cytotoxicity

In the present investigation, the in vitro cytotoxicity of QER-SOLN gel, blank TFS gel (no drug), and QER-TFS gel were evaluated using MTT assay on PCS-200-014 cells (human oral epithelial cell line), as described previously [54]. The cells were cultured in RPMI-1640 medium containing gentamycin (50 µg/mL) and inactivated fetal calf serum (10%). In a humidified atmosphere with 5% CO_2_, the cells were maintained at 37 °C. The cell lines were spread out in the media on 96-well tissue culture plates at a density of 5 × 10^4^ cells per well and then incubated for 24 h. The 96-well plates were subsequently loaded with the tested formulations. After 24 h of incubation, the MTT assay was employed to measure the number of viable cells. In brief, the media was removed and replaced with fresh RPMI 1640 medium (100 µL) without phenol red. Then, MTT stock solution (12 mM, 10 µL) was added to each well, including the untreated controls. The plates were then incubated (37 °C, 5% CO_2_, 4 h). 50 µL of DMSO was added after 85 µL of the medium had been removed from each well, which was then mixed and incubated at 37 °C (10 min). The viable cell numbers were evaluated using a microplate reader (Sunrise, TECAN Inc., San Jose, CA, USA) to measure the optical density at 590 nm. As previously indicated, the viability % was computed [55]

#### 2.2.12. Statistical Analysis

Each experiment was performed thrice; the data are shown as the mean ± SD. In all conducted experiments, except the CCD optimization experiments, one-way ANOVA with the Tukey post hoc test was employed to determine if there were significant differences between the study groups. A *p*-value < 0.05 was considered significant.

## 3. Results and Discussion

### 3.1. Formulation of QER-TFS

Preliminary experiments were carried out to carefully choose the appropriate components and procedure for producing the QER-TFS. The variables were created to provide the utmost encapsulation possible in a suitable VS. These initial studies included the selection of the appropriate solvent, EAs, and sonication time. The approach of film hydration was employed. SDC and Tween 80 provided the highest entrapment and smallest size among several EAs investigated; hence, they were chosen. In order to fabricate transparent and continuous films, a chloroform/methanol combination was selected to dissolve QER and other ingredients utilized in the formulation of TFS. TFS were prepared without sonication and after sonication at various times. For the optimum VS, ten minutes were proven to be adequate.

### 3.2. Experimental Design

The lack of fit was insignificant for all response variables; therefore, the models adequately characterized the range of the estimated results (Table 2). In addition, visual assessment of the model diagnostic plots of dependent variables reveals an excellent fit to the estimated results, with most residual errors following a normal distribution and no noticeable residual error patterns (Figure 1). The link between independent and dependent criteria is illustrated in Figure 2. Table 2 depicts the derived equations, in terms of coded values, that describe the relationship between the independent and dependent variables. 

#### 3.2.1. Effect of Causal Factors on VS

According to Table 1, the average size of QER-TFS vesicles varied from 125.3 to 315.3 nm. When the data of VS of different QER-TFS formulations were evaluated using ANOVA, the linear model was suggested. In addition, the ANOVA analysis revealed that the influence of lipid, EAs, and EAs type on the size of nanovesicles was significant (*p* < 0.05), as stated in Table 2. 

As seen in the first column of Figure 1, increasing the quantity of lipids and EAs led to a decrease in VS. At high lipid concentrations, a modest VS can only be maintained if the EA concentration is raised proportionally. A greater quantity of lipids and EAs promotes better surface coverage of the nanovesicles, hence reducing the interfacial tension, which enables the formation of small nanovesicles [30]. In addition, formulations including SDC exhibited the lowest VS values compared to those containing Tween 80. These results might be explained by the reality that incorporating SDC (charged molecules) into the vesicle bilayer encourages steric repulsion between charged particles thus increasing the curvature of the membrane of nanovesicles and decreasing VS [56]. When the quantity of lipids and EAs exceeded 190 mg and 25 mg, respectively, the lowest VS was observed. 

#### 3.2.2. Effect of Causal Factors on EE

According to Table 1, the EE of produced QER-TFS varied from 51.4% to 86.2%. ANOVA demonstrates that the reduced quadratic model was recommended to analyze gathered entrapment results. In addition, the ANOVA analysis of the collected entrapment data revealed a significant (*p* < 0.05) influence of lipid, EAs, and EAs type on the entrapment of QER.

Increasing lipid and lowering EA seems to enhance the amount of QER entrapped (Figure 1, second column). High phospholipid concentration creates greater space to add additional lipids, reducing the likelihood of the medication leaking to the outer phase [57]. The growth of vesicles rises when EA is incorporated in small amounts [58]. However, an increase in the levels of EA results in the creation of pores in lipid bilayers. Once EA concentration reaches 25 mg, mixed micelles are generated with TFS, resulting in low entrapment due to the rigidity and diminutive size of mixed micelles compared to TFS [59]. Further, the second column of Figure 1 indicates that the entrapment in the case of Tween 80 (HLB:15) was somewhat greater than that of SDC (HLB:16.7). High HLB surfactants may result in drug solubilization in the aqueous media [60].

#### 3.2.3. Effect of Causal Factors on Q_24_

Table 1 demonstrates that the Q_24_ of produced QER-TFS vesicles varied from 49.4 to 84.9%. The reduced quadratic model was suggested when evaluating the Q_24_ results using ANOVA. In addition, the ANOVA demonstrates that the effects of lipid, EAs, and EAs type on Q_24_ of QER-TFS are statistically significant (*p* < 0.05).

Numerous strategies for increased nasal delivery by TFS have been proposed [61,62]. The most probable hypothesis implies that intact TFS may traverse the nasal mucosa and load vesicle-bound medication molecules [63]. The TFS flexibility allows them to enter the nasal mucosa readily. Extreme flexibility enables TFS to traverse pores considerably narrower than their own diameter [64]. Another hypothesis posits that TFS, like other lipid-based nanovesicles, may serve as permeation enhancers [43,65]. In this context, the bilayers of the vesicle traverse the intranasal mucosa, opening “new holes” in the paracellular tight junctions [66], allowing medicines to enter deeper on their own [67].

The highest Q_24_ values were obtained when lipid levels were more than 180 mg (Figure 1, third column). This might be because vesicle phospholipid bilayers have a high affinity for the intercellular lipid layer of the mucosal surface [60]. In addition, TFS containing high (30 mg) or low (10 mg) concentrations of EAs were ineffective for vesicular delivery, but TFS containing a moderate level of EAs (20–25 mg) increased the delivery of QER via the nasal mucosa (third column, Figure 1). As was previously mentioned, vesicles with a large diameter are produced when EA levels are low, whereas mixed micelles, produced when EA levels are high, are less deformable and have poor permeability compared to the TFS system [59,68]. In addition, the greatest Q_24_ was found in nanovesicles containing SDC, as seen in the third column of Figure 1. These may be explained by the SDC’s lower VS, which provided the most surface area for drug diffusion [60]. 

#### 3.2.4. Formulation Optimization

The software suggested an optimal QER-TFS formulation with an overall desirability of 0.80 after imposing constraints on the response variables. As depicted in Table 3, the optimum casual factors were estimated to be 200 mg of lipids, 23.1 mg of EAs, and SDC as the EAs type. The estimated optimal response parameters were VS of 162.6 nm, EE of 76.5%, and Q_24_ of 80.6%. As demonstrated in Table 3, our models effectively predicted the optimum TFS features as the prediction error results for the studied response variables was less than 6%. 

The Pareto chart (Figure 3) shows the impacts of the formulation factors on the response variables. EAs level was the most influential on transferosomal size and cumulative amount permeated (24 h). However, lipid amount was the most influential for encapsulation efficiency.

### 3.3. Characterization of Optimized QER-TFS 

#### 3.3.1. FT-IR 

The FT-IR assessments demonstrated the interaction between QER and TFS matrix (expressed as bond formation or peak absence). Figure 4 illustrates the FTIR spectra of QER, lecithin, SDC, empty TFS (without drug), and QER-TFS. QER has characteristic peaks such as those at 1100–1600 cm^−1^ (aromatic bonding and stretching), 3407 cm^−1^ (stretching phenol O-H), 1663 cm^−1^ (Ketone carbonyl stretching), 1609 cm^−1^ (stretching -C-O), 1521 cm^−1^ (Stretch asymmetrically -C-C=C), 1383 cm^−1^ (-C-OH stretch), and 1261 cm^−1^ (-C-O-C bond) [69]. The characteristic peak of lecithin is 2926 cm^−1^ due to the presence of the carboxylic acid group [46]. SDC showed a distinct peak at 1567 cm^−1^ (Asymmetric COO- stretched band), 2934 cm^−1^ (CH band stretching), 1403 cm^−1^ (symmetric COO- stretched band), and 1042 cm^−1^ (Stretching band for secondary alcoholic C-O) [70]. Empty TFS (no drug) showed similar peaks to QER-TFS. The absence of QER characteristic peaks in QER-TFS spectra suggests that QER molecules were embedded in the nanovesicles.

#### 3.3.2. DSC

Utilizing the DSC apparatus, the degree of nanovesicle crystallinity and the impact of QER integration was investigated (Figure 5). QER displayed a strong endothermic peak at 126 °C, suggesting its crystalline nature. Two peaks of lecithin were detected at 83.5 °C and 154 °C. Moreover, SDC showed an exothermic peak at 189 °C but two endothermic peaks at 102.6 °C and 234 °C. A small endothermic peak was seen in empty TFS at 266 °C. The QER-TFS showed a single peak at 268 °C. The SDC endothermic peak of Empty TFS shifts from 234 to 266 °C on the DSC thermogram, and the lecithin peak disappears [38]. Lecithin peak loss might be triggered by SDC fluidizing the lipid bilayer and producing TFS [71,72]. The physical interaction between SDC and lecithin could shift the SDC endothermic peak [38]. The QER-TFS DSC thermogram is distinguished by relocating the SDC peak to 268 °C and the absence of the QER and lecithin peaks. The lack of the distinctive peak of QER indicates the presence of the drug inside the transferosomal lattice and confirms its amorphous nature [38,73,74].

#### 3.3.3. Morphological Evaluation and Zeta Potential

TEM images of the prepared QER-TFS vesicles (Figure 6) showed vesicles with spherical shape and comparable average VS range similar to the data obtained by using a Zetasizer particle size analyzer. The dispersion is expected to be stable when their surface charge is more than 30 mV (absolute value) [29]. Thereby, the high negative charge value (−32.6 mV) of QER-TFS suggests their high stability.

#### 3.3.4. In Vitro Release

The in vitro release comparison of QER-TFS gel and QER-SOLN gel is shown in Figure 7. The cumulative QER amount released was plotted as a function of time, and as expected, the QER-SOLN gel demonstrated rapid QER release, with a cumulative release of more than 95% within 3 h. In comparison, the release of QER from QER-TFS gel was 61.3% within 3 h. The figure shows bi-phasic drug release behavior from QER-TFS gel, with a rapid drug release occurring in the first 2 h (45.5%) and a slower release over an 8 h period.

#### 3.3.5. Ex Vivo Permeability

The actual drug permeation properties might be revealed by performing the experiments on a natural membrane-like human intranasal mucosa. The sheep nasal mucosa was used in the permeation experiment because of the histological similarities between the sheep and human nasal epithelium [75]. Transferosomal nanovesicles significantly facilitated the increase in the permeability of drug molecules. The permeation profile of QER from optimized QER-TFS gel and QER-SOLN gel is shown in Figure 8. Within 12 h, 77.2 ± 2.6% of QER was permeated from QER-TFS gel, and 46.4 ± 5.2% of the drug was permeated from QER-SOLN gel. The results of this experiment show that QER-TFS gel had a greater permeation rate than QER-SOLN gel (*p* < 0.05). Table 4 summarizes the permeability coefficient and the steady-state flux of QER-TFS gel. The Jss of QER-TFS gel and QER-SOLN gel were 44.35 ± 5.3 and 21.69 ± 2.6 µg cm^−2^ h^−1^, respectively. Collectively, QER-TFS gel formulation resulted in high QER penetration over the nasal mucosa.

#### 3.3.6. Stability Study

The optimized formulation exhibited excellent physical stability throughout a three-month storage period at 4 °C, exhibiting no aggregation, separation, or drug precipitation. Figure 9 depicts the VS, ZP, and EE changes of the stored optimized QER-TFS gel formulation. During the storage period, the preserved samples had no significant differences in VS, ZP, and EE (*p* > 0.05). The high negative ZP originating from the anionic nature of SDC [76] and the nanoscale of the optimal formulation may be the causes of their high stability.

#### 3.3.7. Evaluation of pH

It is necessary to evaluate the pH of QER-TFS gel to guarantee that the system is safe for intranasal administration. In its typical physiological condition, the pH of nasal mucosa varies from 4.5 to 6.5, while the pH of QER-TFS gel was measured to be 6.1. These findings indicate that the QER-TFS gel is physiologically compatible and is not anticipated to cause tissue damage.

### 3.4. Determination of QER Localization in the Brain

CT scanning was used to assess the cellular absorption of the proposed QER-TFS gel in the brain after intranasal application. CT images of rat brains in control, QER-GNPs gel (IN), QER-GNPs (oral), and QER-TFS-GNPs gel (IN) groups are shown in Figure 10. By comparing produced CT scans, we can clearly identify light brain areas that show GNP buildup (yellow arrows). These bright areas may be recognized from nearby tissues because of the considerable X-ray attenuation caused by GNPs. CT scans of QER-GNPs following oral and intranasal administrations (Figure 10B,C) were highly comparable despite the much larger QER dose used orally (300 vs. 10 mg/kg). This finding confirms the adequacy of the intranasal route as an alternative to the most common route of administration, the oral route. The CT scans of the brains of QER-TFS-GNPs gel-treated rats had the maximum intensity (Figure 10D), indicating a higher accumulation of GNPs, corresponding to the previously described findings of the ex vivo permeation investigation via the nasal epithelium, in addition to potential direct passage of the nanotransferosomes to the brain via the trigeminal pathway (i.e., the NTB direct pathway) [30]. The CT experiment hereby demonstrates that the designed intranasal nanoformulations effectively targeted the brain for QER delivery.

### 3.5. In Vitro Assessment of Cytotoxicity

The cytotoxicity test was conducted to ensure the safety of the formulated QER-TFS gel on the epithelial cells. Following intranasal delivery, nanomaterials come in contact with nasal epithelial cells, which constitute the initial barrier to the brain [77]. The PCS-200-014 cells were used as a model for nasal mucosa cells in order to replicate these circumstances and assess the safety and uptake of nanomaterials. As shown in Figure 11, exposure to QER-SOLN gel for 24 h induced significant cell death in PCS-200-014 cells (*p* < 0.05) in contrast with blank TFS gel (no drug) and QER-TFS gel with increasing concentrations of QER. The diminished cytotoxic effect of the transferosomal gel components on the PCS-200-014 cell line demonstrated the suitability of these nanovesicles for biological application and reduced the risk of adverse effects due to the biocompatibility of the nanovesicle constituents.

## 4. Conclusions

This study presents a new delivery system for QER based on transferosomal nanovesicles in an intranasal gel formulation. The prepared formulation had desirable surface charge, pH, and particle size. In comparison to quercetin solution gel, the developed system displayed extended drug release and enhanced drug penetration through the nasal mucosa of sheep. CT scanning of rat brains demonstrated the buildup of GNPs in the brains of all treatment groups, with a greater level of GNPs noted in the rats given the transferosomal gel. Additionally, in vitro studies on PCS-200-014 cells revealed minimal cytotoxicity of QER-TFS in situ gel. Based on these results, the developed transferosomal nanovesicles may be a suitable nanocarrier for QER brain targeting conducted intranasally. 

## Figures and Tables

**Figure 1 pharmaceutics-15-01805-f001:**
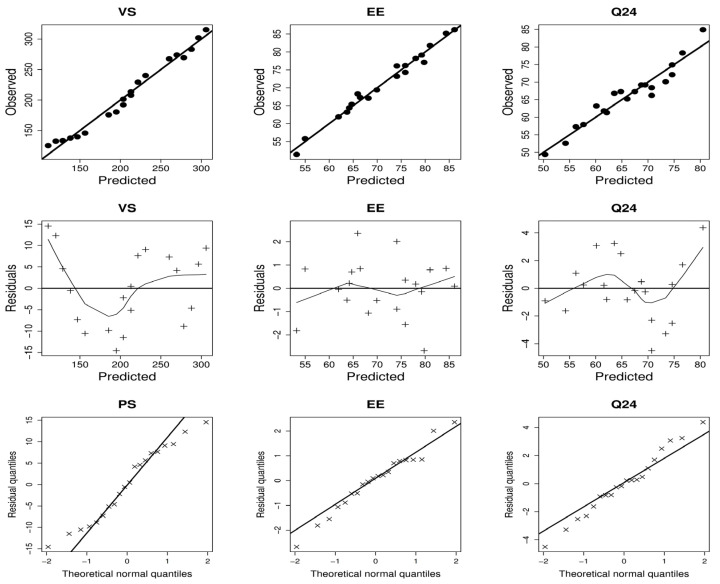
The diagnostic graphs of the response variables models (observed vs. predicted, residuals vs. predicted, and residuals quantiles vs. theoretical normal quantiles).

**Figure 2 pharmaceutics-15-01805-f002:**
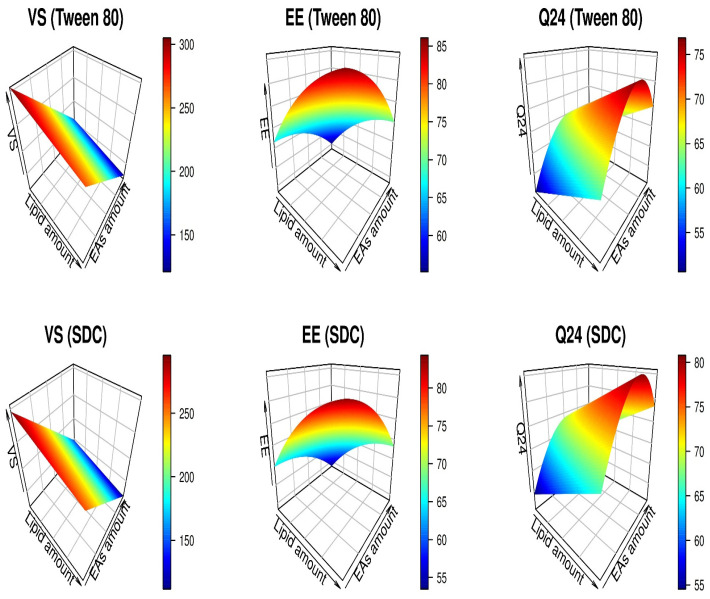
3D plots for the effect of independent variables (Lipid amount, EAs amount, and EAs type) on QER-TFS VS (nm), EE (%), and Q_24_ (%).

**Figure 3 pharmaceutics-15-01805-f003:**
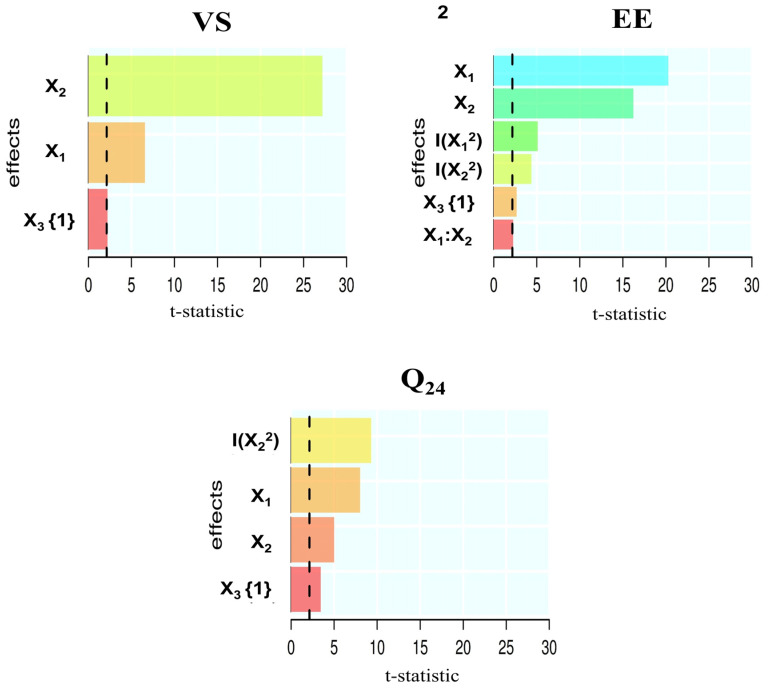
The Pareto chart depicting the impacts of independent variables; Lipid amount (X_1_), EA amount (X_2_), and EA type (X_3_) on response variables.

**Figure 4 pharmaceutics-15-01805-f004:**
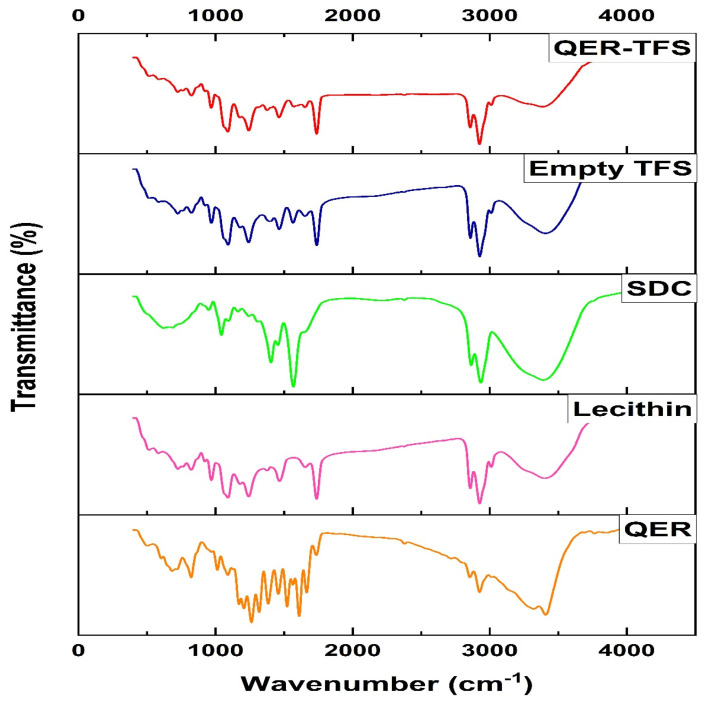
Fourier transform-infrared spectra of QER, Lecithin, SDC, empty TFS (no drug), and QER-TFS.

**Figure 5 pharmaceutics-15-01805-f005:**
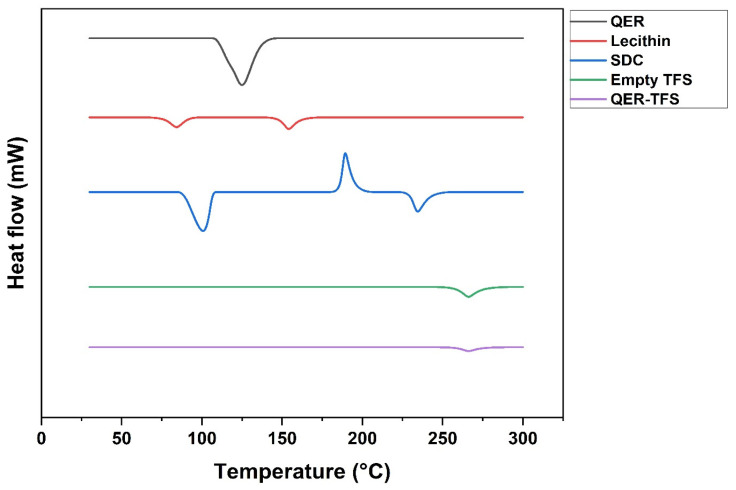
Differential scanning calorimetry thermograms of QER, Lecithin, SDC, empty TFS (no drug), and QER-TFS.

**Figure 6 pharmaceutics-15-01805-f006:**
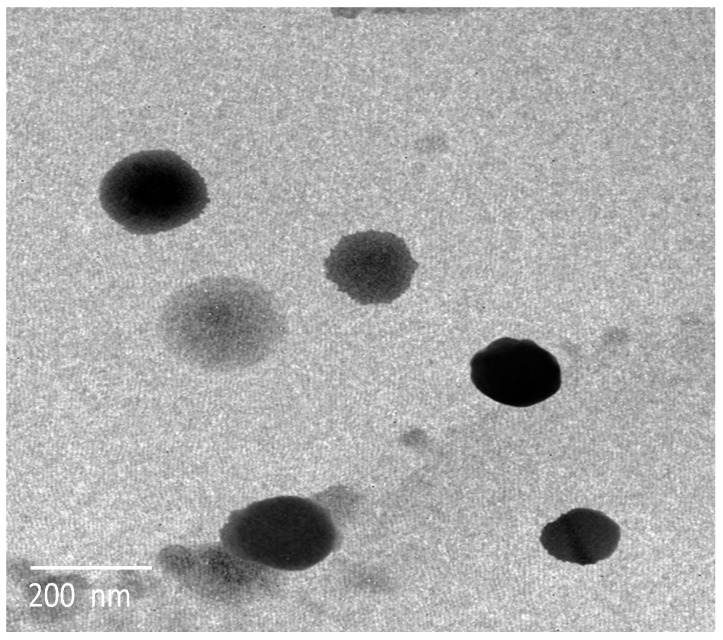
Representative TEM image of QER-TFS dispersion.

**Figure 7 pharmaceutics-15-01805-f007:**
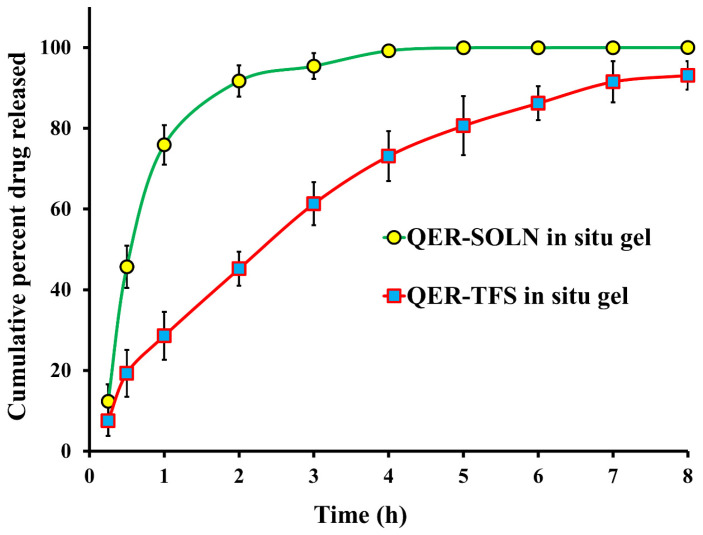
The release profile behavior of QER from optimized QER-TFS in situ gel and QER-SOLN in situ gel through dialysis membranes.

**Figure 8 pharmaceutics-15-01805-f008:**
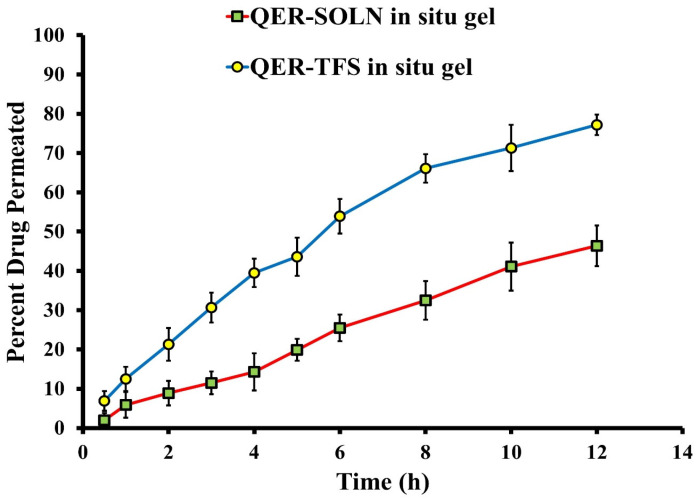
The permeation profile of QER from optimized QER-TFS in situ gel and QER-SOLN in situ gel across nasal membranes.

**Figure 9 pharmaceutics-15-01805-f009:**
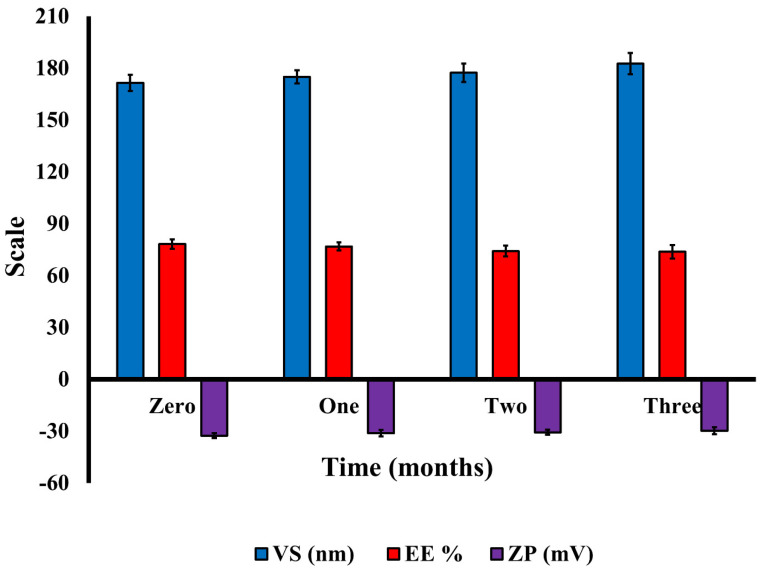
The effect of storage period on the VS, EE, and ZP of QER-TFS gel formulation.

**Figure 10 pharmaceutics-15-01805-f010:**
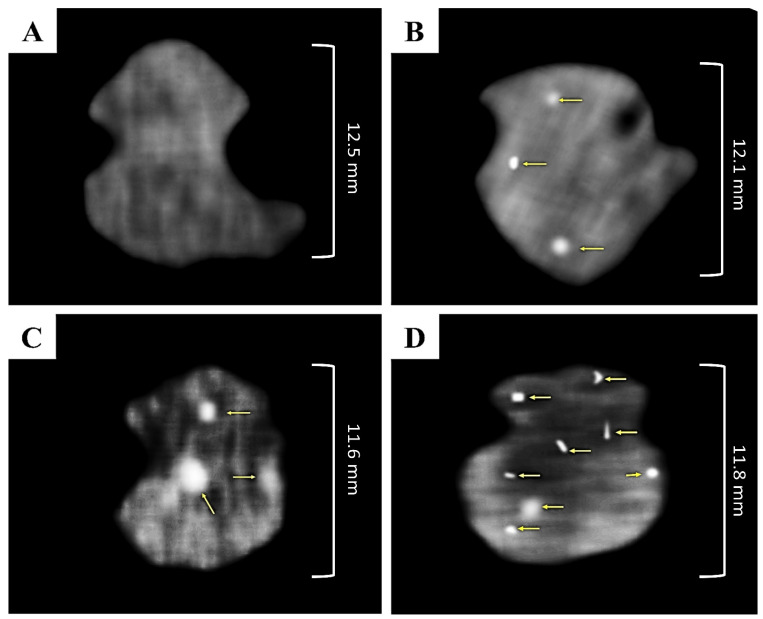
CT images representing cellular uptake of GNPs in the brain (yellow arrow) in (**A**) control group, (**B**) intranasal QER-GNPs gel treated group, (**C**) oral QER-GNPs group, and (**D**) intranasal QER-TFS-GNPs gel treated group.

**Figure 11 pharmaceutics-15-01805-f011:**
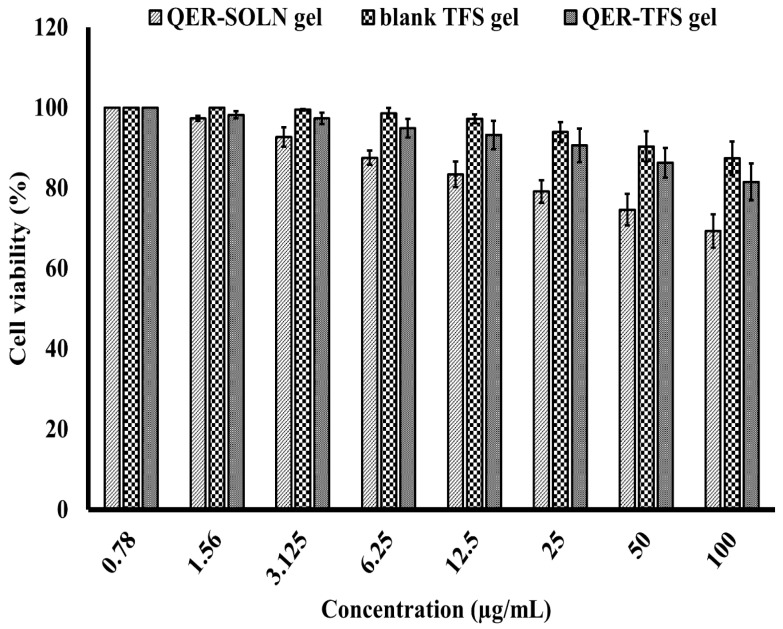
In vitro cytotoxicity of QER-SOLN gel, blank TFS gel (no drug), and QER-TFS gel on PCS-200-014 cells.

**Table 1 pharmaceutics-15-01805-t001:** The levels of independent variables, experimental runs, and response variables for formulating QER-TFS.

Independent Variables			Level			
			−1		0	1
X_1_: Lecithin amount (mg)			100		150	200
X_2_: Edge activator amount (mg)			10		20	30
X_3_: Edge activator type			Tween 80			SDC
Run	X_1_	X_2_	X_3_	Y_1_: VS (nm)	Y_2_: EE (%)	Y_3_: Q_24_ (%)
1	−1	0	−1	240.2 ± 4.1 ^¥^	63.2 ± 4.6	67.3 ± 6.1
2	−1	−1	1	302.1 ± 5.2	65.4 ± 3.2	52.6 ± 5.2
3	0	0	−1	213.5 ± 3.8	74.3 ± 5.1	66.2 ± 5.7
4	1	−1	1	267.6 ± 5.4	85.2 ± 4.8	65.2 ± 4.8
5	1	1	1	125.3 ± 2.6	67.1 ± 3.4	70.1 ± 6.1
6	0	1	−1	137.7 ± 3.9	68.3 ± 4.9	66.8 ± 3.2
7	1	0	−1	180.4 ± 1.6	81.8 ± 3.1	78.3 ± 5.3
8	1	−1	−1	273.9 ± 4.9	86.2 ± 5.3	61.3 ± 4.2
9	−1	−1	−1	315.3 ± 3.1	67.3 ± 4.6	49.4 ± 3.7
10	0	−1	−1	283.2 ± 5.3	77.1 ± 3.5	57.3 ± 4.9
11	0	0	−1	207.9 ± 5.7	76.2 ± 3.6	68.4 ± 5.4
12	1	1	−1	132.5 ± 1.7	69.4 ± 3.7	69.2 ± 4.3
13	1	0	1	175.7 ± 2.4	79.1 ± 2.8	84.9 ± 5.2
14	−1	1	−1	145.8 ± 2.6	55.8 ± 3.9	57.9 ± 3.7
15	−1	1	1	139.6 ± 3.7	51.4 ± 2.9	61.8 ± 2.8
16	0	0	1	192.1 ± 2.8	73.2 ± 3.4	72.1 ± 3.4
17	0	−1	1	269.5 ± 3.7	78.2 ± 2.8	63.2 ± 4.5
18	0	1	1	133.4 ± 2.2	64.4 ± 4.6	67.3 ± 5.1
19	−1	0	1	229.3 ± 3.6	61.9 ± 3.2	69.2 ± 5.6
20	0	0	1	201.4 ± 3.9	76.1 ± 3.8	74.9 ± 4.9

^¥^ data presented as mean ± SD.

**Table 2 pharmaceutics-15-01805-t002:** Data of Regression Analysis and ANOVA of All Dependent Variables.

Source	Size (nm)		EE (%)		Q_24_ (%)	
	F	*p*-Value	F	*p*-Value	F	*p*-Value
Model	262.29	<0.0001	123.41	<0.0001	46.91	<0.0001
X_1_: Lipid amount (mg)	43.17	<0.0001	411.21	<0.0001	64.31	<0.0001
X_2_: Edge activator amount (mg)	738.80	<0.0001	262.92	<0.0001	24.95	0.0002
X_3_: Edge activator type	4.91	0.0416	7.09	0.0195	11.83	0.0037
X_1_X_2_			5.06	0.0425		
X_1_²			26.07	0.0002		
X_2_²			19.14	0.0008	86.53	<0.0001
Lack of Fit	3.38	0.2516	0.6769	0.7298	2.21	0.3543
Model	Linear		Reduced Quadratic		Reduced Quadratic	
%CV	4.57		2.08		3.85	
Adequate precision	45.7876		37.6271		23.7120	
Adjusted R^2^	0.9763		0.9748		0.9062	
SD	9.53		1.48		2.55	
Predicted R^2^	0.9672		0.9602		0.8743	
R^2^	0.9801		0.9827		0.9260	
$VS=208.3-18.1\cdot X_{1}-74.8\cdot X_{2}-4.7\cdot X_{3}$						
$EE=74.97+8.65\cdot X_{1}-6.92\cdot X_{2}-0.88\cdot X_{3}-1.18\cdot X_{1}\cdot X_{2}-3.49{\cdot X}_{1}^{2}-2.99{\cdot X}_{2}^{2}$						
Q_24_$=72.66+5.9\cdot X_{1}+3.68\cdot X_{2}+1.96\cdot X_{3}-10.82\cdot X_{2}^{2}$						

**Table 3 pharmaceutics-15-01805-t003:** Composition, measured, and predicted characteristics of the optimal QER-TFS formulation.

Factor	Optimal	Response	Measured	Predicted	Prediction Error (%) *
X_1_: Lipid amount (mg)	200	VS (nm)	171.4	162.6	5.1
X_2_: Edge activator amount (mg)	23.1	EE%	78.2	76.5	2.2
X_3_: Edge activator type	SDC	Q_24_%	77.2	80.6	4.4

* Calculated as (measured-predicted)/measured × 100.

**Table 4 pharmaceutics-15-01805-t004:** The permeation parameters of QER-TFS versus QER-SOLN.

Formulation	Flux Jss (µg cm^−2^ h^−1^)	Cumulative QER Permeated at 12 h (μg/cm^2^)	Permeability Coefficient (cm/h)
QER-TFS	44.35 ± 5.3	926.4 ± 78.9	0.044347 ± 0.00072
QER-SOLN	21.69 ± 2.6	556.8 ± 156.4	0.021696 ± 0.00047

## Data Availability

Data sharing is contained in this article.

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
