# Peer review of "Intranasal Nanotransferosomal Gel for Quercetin Brain Targeting: I. Optimization, Characterization, Brain Localization, and Cytotoxic Studies"

_pharmaceutics, 2023, doi:10.3390/pharmaceutics15071805_

Round 1
Reviewer 1 Report
The study provides new and interesting data on using transferosomes to deliver neuroprotection agents to the brain via intranasal infusion. Quercetin (flavonoid with multiple neuroprotective effects) was used as a filler for transferosomes. By the combination of FT-IR, DSC, zeta potential analysis, and other methods authors perform optimization of transferosomal lipid composition to achieve better entrapment efficiency and permeation through the nasal mucosa. On the model of PCS-200-014 cells, authors prove the low cytotoxicity of transferosomes themselves.
I have only minor comments on the text:
1) There is a sort of a contradiction between statements on line 64 (QER has the potential to cross BBB) and line 71 (low brain permeation).
2) I do not exactly understand what do authors mean by "acceptable levels" in the phase on line 511.
3) The next phrase on line 513: "system displayed extended drug release and enhanced drug penetration" as compared to what?
Author Response
- The authors would like to express their gratitude to the reviewer for his insightful remarks. Your expertise is much appreciated.
- There is a sort of a contradiction between statements on line 64 (QER has the potential to cross BBB) and line 71 (low brain permeation).
- Authors: We thank the reviewer for the in-depth analysis. The sentences were edited in the manuscript to “However, QER has low distribution potential to the brain after oral delivery owing to difficulties in crossing BBB [16] as well as to rapid metabolism [17]. Therefore, QER exerts its neuroprotective effects several hours after delivery [18]”.
- I do not exactly understand what do authors mean by "acceptable levels" in the phase on line 511.
- Authors: The sentence is now edited to be clearer. We edited it to “The prepared formulation had desirable surface charge, pH, and particle size.”. We thank the reviewer for the kind comment.
- The next phrase on line 513: "system displayed extended drug release and enhanced drug penetration" as compared to what?
- Authors: We performed the in vitro release and ex vivo permeation for QER-TFS gel and QER-SOLN gel. So, we edited the text to “In comparison to quercetin solution gel, the developed system displayed extended drug release and enhanced drug penetration through the nasal mucosa of sheep”. Thank you for your remarks.
Reviewer 2 Report
The author should revise the manuscript as following suggested points
1. Author should draw a schematic illustration that can explain the whole store to the readers at a glance, The illustration keep as Figure 1
2. In Figure 4 FT-IR spectra the baseline is not proper author should revise it to make the smooth baseline of each spectrum and represent it with proper labelling of significant peaks.
3. DSC result explain in detail
4. In Figure 7 the QER-SOLN why error bar is not in some of the points?
5. In Figure 8 why error bar is not only on the positive side but not on the negative side is there any special reason behind it if not include both?
6.In the figure 9 what is in the (-) Y axis Zero, One , Two and Three
explain in the text
7. Figure 10 CT images are not clear replace them with high-quality CT images.
8. In support of Figure 11 cytotoxicity test include high-quality cells image of each group.
9. in the cytotoxicity assay 0.78 microliter/ml group do not have error bar
10 A high-quality QER-TFS gel picture included in figure 1.
11. In the introduction section or discussion section compare your out come with other carriers such as micelles Synthesis and multi‐responsive self‐assembly of cationic poly (caprolactone)–poly (ethylene glycol) multiblock copolymers. Chemistry–A European Journal, 23(34), 8166-8170. and explain how your system is better than other systems and cite the following article and above as well, Therapeutic Efficacy of Herbal Formulations Through Novel Drug Delivery Systems. In Enhancing the Therapeutic Efficacy of Herbal Formulations (pp. 1-42). IGI Global. in the discussion section of this manuscript compare with other data clearly.
NA
Author Response
- The authors would like to express their gratitude to the reviewer for his insightful remarks. Your expertise is much appreciated.
- Author should draw a schematic illustration that can explain the whole store to the readers at a glance, The illustration keeps as Figure 1.
- Authors: Thank you for your recommendation. We made a graphical abstract that summarizes the work in the manuscript.
- In Figure 4 FT-IR spectra the baseline is not proper author should revise it to make the smooth baseline of each spectrum and represent it with proper labelling of significant peaks.
- Authors: Figure 4 is now edited to show the baseline of each spectrum. The characteristic peaks of each component have been mentioned in the text of the manuscript. Thank you so much.
- DSC result explain in detail.
- Authors: The main features of the DSC thermogram are the melting point and the degree of crystallinity, and both features were discussed thoroughly in the results and discussion section and compared between the QER and TFS matrix. The Authors thank the reviewer for the in-depth analysis.
- In Figure 7 the QER-SOLN why error bar is not in some of the points?
- Authors: At some points, the dispersion (calculated as standard deviation) is so narrow relative to other points. So, on graph, error bars may not show clearly for the points with narrow dispersion. Thank you.
- In Figure 8 why error bar is not only on the positive side but not on the negative side is there any special reason behind it if not include both?
- Authors: Both positive and negative sides are now included in Figure 8. Thank you so much.
- In the figure 9 what is in the (-) Y axis Zero, One, Two and Three. explain in the text
- Authors: The primary horizontal axis title is now added to Figure 9. Thank you for your suggestion.
- Figure 10 CT images are not clear replace them with high-quality CT images.
- Authors: Figure 10 is now replaced by a higher resolution Figure. Thank you so much.
- In support of Figure 11 cytotoxicity test include high-quality cells image of each group.
- Authors: In the technique used in the cytotoxicity studies, only cell viability is reported with no images. This is usually the case with cytotoxicity studies (please refer to some of the published cytotoxicity studies for confirmation: https://doi.org/10.1016/j.ijbiomac.2018.05.079; https://doi.org/10.1007/s13346-019-00622-5; https://doi.org/10.1016/j.ijpharm.2022.121563).
- In the cytotoxicity assay 0.78 microliter/ml group do not have error bar.
- Authors: At some points, the dispersion (calculated as standard deviation) is so narrow relative to other points. So, on graph, error bars may not show clearly for the points with narrow dispersion. Thank you.
- A high-quality QER-TFS gel picture included in figure 1.
- Authors: A high-quality QER-TFS gel picture is now included in the graphical abstract. Thank you so much.
- In the introduction section or discussion section compare your outcome with other carriers such as micelles Synthesis and multi‐responsive self‐assembly of cationic poly (caprolactone)–poly (ethylene glycol) multiblock copolymers. Chemistry–A European Journal, 23(34), 8166-8170. and explain how your system is better than other systems and cite the following article and above as well, Therapeutic Efficacy of Herbal Formulations Through Novel Drug Delivery Systems. In Enhancing the Therapeutic Efficacy of Herbal Formulations(pp. 1-42). IGI Global. in the discussion section of this manuscript compare with other data clearly.
- Authors: In the introduction section, we already pointed out the benefits of using transferosomal nanovesicles as carriers for brain delivery relative to other carriers. The paragraph is highlighted with yellow color. The second article suggested by the reviewer “Therapeutic Efficacy of Herbal Formulations Through Novel Drug Delivery Systems. In Enhancing the Therapeutic Efficacy of Herbal Formulations (pp. 1-42). IGI Global” is now added to the aforementioned paragraph (Reference 7).
- Concerning the first article suggested by the reviewer “Synthesis and multi‐responsive self‐assembly of cationic poly (caprolactone)–poly (ethylene glycol) multiblock copolymers. Chemistry–A European Journal, 23(34), 8166-8170”, the authors feel that it is irrelevant to the topic in hand as it is focused on the chemical synthesis of novel carriers without testing the potentials of the synthesized carriers as delivery systems to the brain via the nose-to-brain route. So, the article was not included. Thank you for your recommendation.
Reviewer 3 Report
1, There is very limited basis for the statement “Based on these results, the developed transferosomal nanovesicles are a suitable nanocarrier for QER brain targeting through the intranasal route with the potential for improved management of neurological disorders.” What is the efficiency for brain targeting? What is the disease model used as neurological disorders? If I understand correctly, this manuscript could be the part I of a whole research? In that case, the authors should combine the biological evaluation together into a single piece, and some unnecessary characterizations should be compressed or moved into supporting information.
2, The authors mainly focused on the composition optimization and characterization of the gel, while the application is off the table. This is not in accordance with the title.
3, What is the principle for brain delivery via the intranasal route (size, zeta-potential, density…)? How did the system match the principle?
4, What is PCS-200-014 cell model? Are there any supporting references for this human oral epithelial cell line used as a model for nasal mucosa cell?
5, The detailed brain accumulation of the aimed drug should be calculated based on a reliable animal protocol.
Author Response
- The authors would like to express their gratitude to the reviewer for his insightful remarks. Your expertise is much appreciated.
- There is very limited basis for the statement “Based on these results, the developed transferosomal nanovesicles are a suitable nanocarrier for QER brain targeting through the intranasal route with the potential for improved management of neurological disorders.” What is the efficiency for brain targeting? What is the disease model used as neurological disorders? If I understand correctly, this manuscript could be the part I of a whole research? In that case, the authors should combine the biological evaluation together into a single piece, and some unnecessary characterizations should be compressed or moved into supporting information.
- Authors: The authors agree with the reviewer that there is no enough evidence in this part of study (Part I) to support our claim that the developed delivery system may improve management of neurological disorders, accordingly, this declaration is now removed and the sentence is now rephrased to be “Based on these results, the developed transferosomal nanovesicles may be a suitable nanocarrier for QER brain targeting through the intranasal route”. However, the study contains enough evidence to support the claim that the developed system is able to improve QER’s accumulation in the brain. CT scanning is a valid and reliable method to determine the ability of the delivery system to target the drug to the brain, and was previously employed elsewhere (see for example: https://doi.org/10.1080/1061186X.2019.1608553; https://doi.org/10.1016/j.xphs.2022.02.012; https://doi.org/10.1039/C3NR04878K). Thank you so much for your in-depth analysis.
- The authors disagree with the reviewer that we should combine the two parts of the study into a single manuscript because of the different objectives. In Part I, the objective was a little bit general; to show a proof-of-concept that transferosomes are a suitable carrier to the brain through the nose and can be an added value for boosting brain accumulation of multipurpose nutraceuticals such as QER. In Part II, the objective was more specific as it was pointed towards testing the system developed in Part I as a potential treatment for a specific neurodegenerative disease, the depression. Dividing research work to integrated parts is not odd and was previously implemented several times in the literature (see for example: https://doi.org/10.1002/jps.23205; https://doi.org/10.1002/jps.23001; https://doi.org/10.1007/s11743-013-1470-4; https://doi.org/10.1007/s00170-006-0585-4; https://doi.org/10.1016/j.ijpharm.2013.03.034; https://doi.org/10.1016/j.jconrel.2009.12.026; https://doi.org/10.1007/s11095-013-1213-2; https://doi.org/10.1007/s11095-005-8343-0; https://doi.org/10.1007/s11051-010-0071-7; https://doi.org/10.1016/j.jconrel.2013.11.023). Thank you so much for your in-depth analysis.
- The authors mainly focused on the composition optimization and characterization of the gel, while the application is off the table. This is not in accordance with the title.
- Authors: The authors agree with the reviewer that the main focus of the manuscript was on the composition optimization and characterization of the gel, and the application was off the table. This is because the main focus of the second part of the study (Part II) was on the application, where we tested our optimized system as a potential treatment for a serious neurodegenerative disease, the depression. Thank you so much.
- What is the principle for brain delivery via the intranasal route (size, zeta-potential, density…)? How did the system match the principle?
- Authors: The items pointed out by the reviewer (size, zeta-potential, density…) are not principles for brain delivery. Nonetheless, they are factors that may significantly contribute to the brain delivery process. However, to address the question raised by the reviewer on how the interplay of these factors has influenced the brain delivery process, a mechanistic study is necessary, which is beyond the scope of this study. Nevertheless, we discussed the findings of previous reports investigating into how brain delivery can benefit from the physical properties of nanoparticles, especially transferosomes (please refer to the highlighted paragraph in the introduction).
- What is PCS-200-014 cell model? Are there any supporting references for this human oral epithelial cell line used as a model for nasal mucosa cell?
- Authors: There are no references for using this cell line as a model for nasal mucosa cells. However, some studies used other kinds of oral human epithelial cell lines as a model for nasal mucosa cells, such as those reported by Elsenosy et al. (DOI: 2147/IJN.S277352). The authors appreciate your comprehensive review.
- The detailed brain accumulation of the aimed drug should be calculated based on a reliable animal protocol.
- Authors: To calculate the accumulation of the drug in the brain of an animal model, a pharmacokinetic study using plasma and brain tissue is necessary. Unfortunately, there is no reliable assay to measure total quercetin (conjugated and nonconjugated) in-vivo since most of the applicable assays measure the parent drug along with its active methylated metabolite (isorhamnetin) (see for example: https://doi.org/10.1016/j.freeradbiomed.2011.06.017; https://doi.org/10.1016/j.nut.2010.09.002; https://doi.org/10.1016/j.jnutbio.2016.02.001; https://doi.org/10.1016/j.phrs.2009.08.006; https://doi.org/10.1016/j.nano.2007.12.001).
As quercetin and its active metabolite are practically inseparable assay-wise, relying on standard pharmacokinetic protocols to quantify brain accumulation of quercetin may have led to erroneous results. Rather, we relied on a qualitative analysis to demonstrate the brain targeting capability of our system. CT scanning is a valid and reliable method to determine the ability of the delivery system to target the drug to the brain, and was previously employed elsewhere (see for example: https://doi.org/10.1080/1061186X.2019.1608553; https://doi.org/10.1016/j.xphs.2022.02.012; https://doi.org/10.1039/C3NR04878K).
Round 2
Reviewer 2 Report
Improve all figure labelling and appearance such as in Figure 3
NA
Author Response
- The authors would like to express their gratitude to the reviewer for his insightful remarks. Your expertise is much appreciated.
- Improve all figure labelling and appearance such as in Figure 3.
- Authors: All Figures are now replaced by higher resolution Figures (Resolution was increased from 300 to 600 dpi). Thank you so much.
Reviewer 3 Report
I could learn from the hints that this could only be a part of a whole work. I finally chose "rejection" as my original decision, based on my understanding of that manuscript as an independant work. I think if the in vivo evaluation has already been done, this work could be worthy being reviewed (but it still depends). However, I still strongly suggest the so-called Part I and Part II work should be combined together as a single piece of work to be published. Neither of the two parts shall be published solely, since there are more than 100 methods in expressing rich information in a limited space.Author Response
- The authors would like to express their gratitude to the reviewer for his insightful remarks. Your expertise is much appreciated.
- I could learn from the hints that this could only be a part of a whole work. I finally chose "rejection" as my original decision, based on my understanding of that manuscript as an independent work. I think if the in vivo evaluation has already been done, this work could be worthy being reviewed (but it still depends). However, I still strongly suggest the so-called Part I and Part II work should be combined together as a single piece of work to be published. Neither of the two parts shall be published solely, since there are more than 100 methods in expressing rich information in a limited space.
- Authors: With all due respect to the reviewer’s opinion regarding the necessity of merging the two parts into a single manuscript, we strongly believe that each part constitutes a standalone study that tells an independent story and reports a different set of experiments to support its own objectives which are completely disparate between the two parts. The first part focuses on development, optimization and in-vitro/in-vivo characterization of a drug delivery system for brain targeting whereas the second part evaluates the therapeutic potential of the optimized system through testing in an animal model of depression via a battery of pharmacological tests.
The authors believe that the reviewer’s decision to “reject” the Part I manuscript on the grounds of incomplete work is “not fair” especially that some RECENT studies addressing the issues of nose-to-brain delivery and brain targeting have relied ONLY on in-vitro experiments (see for example https://doi.org/10.1016/j.carpta.2023.100332) or CT scanning (see for example https://doi.org/10.1080/1061186X.2019.1608553) to declare their developed delivery systems able to target the brain.